# A Longitudinal Study of Sexual Entitlement and Self-Efficacy among Young Women and Men: Gender Differences and Associations with Age and Sexual Experience

**DOI:** 10.3390/bs6010004

**Published:** 2016-01-18

**Authors:** Gillian Hewitt-Stubbs, Melanie J. Zimmer-Gembeck, Shawna Mastro, Marie-Aude Boislard

**Affiliations:** 1School of Applied Psychology and Menzies Health Institute of Queensland, Griffith University, G40 Parklands Dr., Southport 4222, Australia; gillian.hewitt-stubbs@griffithuni.edu.au (G.H.S.); s.mastro@griffith.edu.au (S.M.); 2Département de sexologie, Université du Québec à Montréal, C.P. 8888, Succ. Centre-ville, Montréal, QC H3C 3P8, Canada; boislard-pepin.marie-aude@uqam.ca

**Keywords:** sexual subjectivity, sexual self-perceptions, sexual behavior, gender, young adults, adolescents

## Abstract

Many scholars have called for an increased focus on positive aspects of sexual health and sexuality. Using a longitudinal design with two assessments, we investigated patterns of entitlement to sexual partner pleasure and self-efficacy to achieve sexual pleasure among 295 young men and women aged 17–25 years attending one Australian university. We also tested whether entitlement and efficacy differed by gender, and hypothesized that entitlement and efficacy would be higher in older participants and those with more sexual experience. A sense of entitlement to sexual partner pleasure increased significantly over the year of the study, whereas, on average, there was no change in self-efficacy over time. At Time 1 (T1), young women reported more entitlement than young men. Age was positively associated with T1 entitlement, and experience with a wider range of partnered sexual behaviors was concurrently associated with more entitlement and efficacy and was also associated with increased entitlement to partner pleasure and increased self-efficacy in achieving sexual pleasure at T2 relative to T1. A group with the least amount of sexual experience was particularly low in entitlement and efficacy when compared to groups with a history of coital experience. There was no evidence that any association differed between young men and young women. Limitations of the study include a sample of predominantly middle class, Caucasian students at one university and the possibility that students more interested in sex and relationships, and with more sexual experience, chose to participate.

## 1. Introduction

Sexuality is a “central aspect of being human” [1] (p. 15) and it is often defined to include sexual behavior, the development of sexual preferences, the understanding of the self as a sexual being, sexual decision-making, and agency [2]. It also includes the understanding of others’ sexual desires and how to approach and handle intimacy with another person. The World Health Organization [3] (p. 5) described sexuality as “an integral part of the personality of everyone: man, woman and child. It is a basic need and aspect of being human that cannot be separated from other aspects of life. It influences thoughts, feelings, action and interactions and thereby our mental and physical health”. 

Sexuality can undergo change and evolve throughout much of life. Nevertheless, because adolescence and early adulthood are periods of considerable change and development through an increase in independence and maturing both physically and psychologically [4,5,6,7], this can be the stimulus for significant changes in sexual behavior, as well as attitudes and beliefs about the sexual self and sexual relationships with others [8,9]. Research shows this is a time of defining sexuality [9,10] and experimenting with sexual behavior [11]. For example, research with Australian secondary school students indicates that the majority of students in grades 10 to 12 (approximate ages of 15 to 17 years) have engaged in some form of partnered sexual behavior [12]. In one report [13] a large proportion of Australians (about 65%) reported being sexually active (defined as reporting a history of vaginal sexual intercourse) before they were 18-years-old.

Surprisingly, although there is some earlier writing on normative sexual development [14], it is only in the last decade that sexuality and sexual behavior have been emphasized as normal and expected aspects of adolescent and young adult life [15,16,17,18,19]. More commonly, adolescent sexuality has been considered a risky behavior for undermining good health and future development, and described as a part of a syndrome of problem behavior [11,17,18,19,20]. Given this framing of adolescent sexuality, much of the research has focused on behavior and health correlates, including early onset of sexual intercourse, adolescent pregnancy, sexually transmitted infections (STI) risk, consistency of condom use, and (increasingly) sexual and dating violence in the lives of adolescents [17].

Despite this history, sexual health is now increasingly a focus in contemporary adolescent research, with many scholars continuing to call for an increased focus on positive aspects of sexual health and sexuality [1,2,3,8,14,16,17,18,19,20,21,22,23,24,25,26,27]. Sexual health refers to sexual behaviors, sexual relationships and sexual self-conceptions, including those that are risky for well-being and health along with those that may be positive and growth-promoting. For example, sexual health has been described as including low rates of risk-taking behaviors, but is also described as the capacity to “appreciate one’s own body”, to “express love and intimacy”, and to “enjoy and express one’s sexuality throughout life” [21]. One area of support for research on sexual health and positive or typical sexual development comes from the developmental psychopathology perspective, which highlights that the understanding of atypical development or risky behavior pathways must be founded upon a comprehensive understanding of all aspects of typical or normative developmental trajectories [27]. Therefore, in order to understand atypical development, we also require a more comprehensive understanding of all aspects of typical development [28].

### 1.1. Entitlement to Sexual Partner Pleasure and Sexual Self-Efficacy to Achieve Pleasure

#### 1.1.1. Definition and Measurement

The notions of sexual subjectivity [19,24,29,30] and sexual self-concept [28,31,32] have proven useful as ways to consider and assess one's understanding of the self as a sexual being, as an aspect of sexual health and sexuality. However, understanding of the self as a sexual being includes multiple components. Some of these components relate to the cognitions, emotions and sensations associated with sexual development and the inclusion of a sexual self in one’s global self-definition and identity. In some recent research, these dimensions have been captured with the examination of young women's sexual subjectivity, which includes positive feelings of sexual self-esteem, feelings of greater entitlement to and efficacy to achieve pleasure, and capacity for and engagement in self-reflection about sexuality and sexual behavior [30]. This description of sexual subjectivity was founded in existing conceptualizations of girls' sexuality as an aspect of the self as the subject, rather than the object, of sexual desire and pleasure [19,29,33], and has been defined as “perceptions of pleasure from the body and the experience of being sexual” [19] (p. 28). It involves feeling entitled to sexual pleasure, and possessing the agency to make sexual decisions and take ownership over one’s pleasure [23,29]. Tolman [29] pointed out that developing a sense of the self as a sexual subject is necessary for girls and young women in order to make active sexual choices that meet one's needs for both sexual pleasure and sexual safety. Sexual subjectivity develops and becomes an integrated aspect of the self-concept during adolescence, and it aligns well with definitions of sexual self-concept, which often include sexual openness, sexual esteem and low sexual anxiety [31,32].

Horne and Zimmer-Gembeck [19,30] developed and validated the first standardized tool to assess sexual subjectivity, the Female Sexual Subjectivity Inventory (FSSI). After a review of the literature related to themes identified in sexual subjectivity research and theory, sexual subjectivity was operationalized as sexual self-perceptions and feelings about sex, including sexual body esteem, a sense of entitlement to sexual pleasure, self-efficacy in achieving sexual pleasure, and the capacity for and engagement in sexual self-reflection [30]. The first element, sexual body esteem was defined as positive feelings about the body [30,34]. Buzwell and Rosenthal [35] identified that sexual body esteem was linked to self-perceptions of perceived sexual attractiveness and desirability. It relates to an understanding and experience of pleasure with the body [33]. Other elements of sexual subjectivity (sense of entitlement to sexual pleasure; self-efficacy in achieving sexual pleasure) relate to the cognitive and emotional components of sexual pleasure, such as recognizing sexual desires and understanding the experience of pleasure [17,29]. Writing about young women, Tolman [29] theorized that the better a young woman understands her sexual desires and pleasures the more likely she will be able to maintain relationships and protect herself from sexual risks.

The final element of sexual subjectivity is sexual self-reflection, which refers to cognitive and emotional reflection on sexual attitudes and behavior [30,34]. Cyranowski and Anderson [36] demonstrated that the ability to reflect critically on sexual experience and make decisions about future sexual strategies and behavior could be an important component of healthy sexual development. In the present study, we focus on two of these elements of sexual subjectivity that are most related interactions with others, given our focus on partnered sexual behavior and on the age periods of late adolescence and early adulthood, a time when romantic relationships and sexual behavior becomes more prevalent and frequent [37,38,39,40]. These two elements include a sense of entitlement to sexual partner pleasure and self-efficacy to achieve sexual pleasure.

#### 1.1.2. The Development of Sexual Subjectivity

Founding our hypotheses on developmentally-sensitive theories of sexuality (e.g., [29]), entitlement to sexual partner pleasure and sexual self-efficacy were hypothesized to progress with age and sexual experience. Regarding age, cross-sectional [19,41] and longitudinal [34] research studies have suggested that sexual subjectivity develops with age. In one of these studies [19], perception of entitlement to partner pleasure and sexual self-efficacy were higher among females in their early twenties compared to those in their late teens. In a longitudinal study of young women [34], entitlement and efficacy were higher among participants who had a greater range of experience with different sexual behaviors. Interestingly, no associations between age and entitlement or efficacy were found in this longitudinal study. Such associations between sexual subjectivity, age, and sexual behavior have not yet been examined in young men. Yet, although not directly focused on sexual subjectivity, other research is suggestive of the role of sexual exploration and experience in promoting greater sexual subjectivity among both young men and women. In a 4-year study, adolescents' sexual experience had a bidirectional relationship to their sexual self-concept [31]. Hence, sexual self-concept became more positive over time in response to sexual behavior gains, and improvements in sexual self-concept also foreshadowed increased sexual behaviors over time.

### 1.2. Gender Differences in Sexual Behavior, Attitudes, and Sexual Subjectivity

Gender differences in sexual behavior have been documented [11,42,43]. A meta-analysis by Petersen and Hyde [43] summarized the findings for adults. In relation to behavior, the most prominent gender difference was in autoerotic behaviors (e.g., masturbation), where men were found to have higher rates of autoerotic behavior than women. In another review, men were found to report an earlier age of first sexual intercourse [44], and a history of more sexual partners and more consistent sexual activity (e.g., having sexual intercourse more often) than women [43,45]. However, most of the gender differences found were statistically small, and within-gender variation was larger than between-gender difference.

Gender differences in emotional reactions to and satisfaction with sex have also been documented [43,46], though less extensively. Using data from the USA National Longitudinal study of Adolescent Health, significant gender differences in the expectations that sexual intercourse will be pleasurable were found, with 30% of girls agreeing with this statement as opposed to 62% of the boys [46]. Ott, Millstein, Ofner, and Halpern-Felsher [47] found that girls valued intimacy significantly more and pleasure significantly less than boys and that girls had lower expectations that sex would meet their goals of pleasure than boys did. Galinsky and Sonenstein [48] also found that women report less enjoyment from partnered sexual interactions than men. This is thought to be because women tend to prioritize male pleasure over their own [49], which may result in personal sexual desires becoming secondary to their partners' desires [50]. Qualitative studies have reported that women have more difficulty than men in understanding their sexual feelings and in communicating their desires effectively [29,51]. Traditionally, there has been stigma attributed to women being upfront and explicit in such communication [52].

It is expected that the gender differences found in sexual behavior and attitudes would also apply to the perception of entitlement to sexual partner pleasure and sexual self-efficacy, as this has been reported in one recent study [53]. In this previous study, young women reported a greater sense of entitlement to sexual partner pleasure, but less self-efficacy in achieving sexual pleasure than did young men. Thus, women may feel *more* entitled to pleasure, but may feel *less* efficacious in achieving pleasure. These findings concur with evidence that women report less enjoyment from partnered sexual interactions than men [48]. Women, especially young women, may feel less efficacious because they feel entitled but are not feeling the pleasure they expect. Partly, this could be due to the known difficulty that girls and women face when attempting to communicate their sexual desires [29,51]. Recent research supports this suggestion [54], showing that increased sexual subjectivity, in the forms of entitlement to sexual partner pleasure and self-efficacy in achieving sexual pleasure, was associated with a greater likelihood of young women's direct communication about their sexual desires.

### 1.3. The Current Study

As the quantitative data accumulate on sexual subjectivity, its role in personal health and well-being has been shown in multiple studies of young women [11,19,34,41], and in one cross-sectional study of young men [53]. To better inform sexual health programs, additional research is needed which includes young men, and also assesses elements of sexual subjectivity and its development over time. In the current study, this research gap was addressed by: (1) investigating differences between young men and women (aged 17 to 25 years) in sense of entitlement to sexual partner pleasure and sexual self-efficacy over 1-year with two waves of data collection; and (2) examining whether entitlement and efficacy were greater among older compared to younger participants and those with more compared to less experience with partnered sexual behaviors.

The following three general hypotheses were tested: (1).*Gender differences*: Sexual subjectivity at Time 1 (T1) and Time 2 (T2) will differ between young women and young men. In particular, as found in the one previous study comparing young women and men on sexual subjectivity [53], it is expected that young women will be higher than young men in a sense of entitlement to sexual partner pleasure and lower than young men in sexual self-efficacy;(2).*Sexual subjectivity as a function of age*: Supporting the theory that sexual subjectivity develops with increasing age during adolescence and young adulthood [29], sexual subjectivity will have a positive association to age at T1 and T2 of this study. Previous cross-sectional studies have found that age was associated with sexual subjectivity [19];(3).*Sexual subjectivity as a correlate of total sexual experience*: Based on earlier research on the association of sexual behavior with greater sexual subjectivity [34,53], sexual subjectivity will be associated with more sexual experience at both T1 and T2. In addition, greater T1 sexual experience will be associated with greater increases in sexual subjectivity at T2 relative to T1.

## 2. Method

### 2.1. Participants

The participants were 295 adolescent and young adult men (*n* = 112) and women (*n* = 183) aged between 17 to 25 years (*M* = 19.5 years, *SD* = 1.9). Although an exact count was not available, we estimate that 375 students were approached to participate, resulting in a participation rate of approximately 79%. Participants completed two waves of data collection approximately 1-year apart. Overall, 91% were white/Caucasian, 3% were Asian, 1 was Aboriginal/Pacific Islander, and the remaining participants indicated an “Other” sociocultural background. Most lived with their parents (61%) and reported being only attracted to the other sex (80%). At T1, 225 of the participants (76%) reported a history of sexual intercourse and 9% reported having had no sexual experience apart from kissing. 

Of the 295 participants, 179 participants completed the second wave of data collection (T2). To maintain all 295 participants in all analyses, multiple imputation was used to estimate T2 missing data. Ten imputed datasets were estimated and pooled results are reported.

### 2.2. Measures

#### 2.2.1. Sexual Subjectivity

Two elements of sexual subjectivity were measured with 8 items (4 items assessed entitlement to sexual partner pleasure and 4 items assessed sexual self-efficacy for achieving partner pleasure) drawn from the Female Sexual Subjectivity Inventory (FSSI) [30]. The same items are used to assess these two elements of sexual subjectivity on the Male Sexual Subjectivity Inventory (MSSI) [53]. Participants answered each item with responses ranging from 1 (*strongly disagree*) to 5 (*strongly agree*). Cronbach’s α for sense of entitlement to sexual partner pleasure was 0.80 at T1 and 0.82 at T2. Cronbach’s α for sexual self-efficacy was 0.82 at T1 and 0.87 at T2. Composite scores were produced by averaging the responses to the items on each of the two subscales. A higher score reflected more entitlement or efficacy.

#### 2.2.2. Sexual Behavior

Participants indicated if they had experienced any of the following sexual activities: romantic kissing (tongue/French), sexual fantasizing, self-masturbation, light touching (above the waist), heavy touching (below the waist), oral sex (giving), oral sex (receiving), sexual intercourse (vaginal penetration). Participants who reported sexual intercourse were asked how old they were the first time. Two composite measures were formed. The first measure (Sexual Behavior Repertoire) was a sum of different sexual behaviors reported by each participant at T1. The second measure (Sexual Behavior Group) was based on the particular sexual behaviors reported at T1, following the procedures used in past research [15,30,34]. Five groups were created. Participants who reported never experiencing coitus were divided into two groups of (1) inexperienced (those who reported no prior sexual experiences involving a partners’ genitalia, such as heavy petting, giving or receiving oral sex, orgasms, *n* = 27); and (2) experienced no coitus (those who reported at least one sexual experience which involved interaction with their partners’ genitals but did not include coitus, *n* = 47); Participants who reported experience with coitus were divided into three groups based upon the age they first had sexual intercourse;(3) first coitus before age 16 (*n* = 60); (4) first coitus at 16 (*n* = 53); (5) first coitus after age 16 (*n* = 108).

### 2.3. Procedure

Approval from the university Human Research Ethics Committee was obtained prior to data collection. At Time 1 (T1), participants were approached at a university campus in Australia in the week before classes commenced (*i.e.*, during orientation week) and asked to participate in a study “About You and Your Relationships”. The front cover of the survey described the questions as focused on personal sexual and romantic experiences, and stressed the confidential nature of the survey.

Participants completed the T1 questionnaire under the supervision of a research assistant. Each participant generated her/his own personal code, which was noted on the questionnaire. This code was designed to be recalled using a series of prompts included on the T2 survey. Contact details for the T2 survey were collected on a separate form to ensure confidentiality. At T2 participants were individually contacted via email or telephone. Once their commitment to proceed with the research had been reconfirmed, they were sent an email with prompts to remind them of their identity code and a link to an online version of the questionnaire. Upon completion of each questionnaire, each participant received a voucher for a coffee.

## 3. Results

### 3.1. Preliminary Analyses

We compared responses on all measures between individuals who specifically identified as being only attracted to the other sex (*n* = 237) and other participants (*n* = 58). There was no difference in sexual subjectivity or sexual behavior repertoire between groups, but a higher proportion of participants who reported they were only attracted to the other sex reported a history of vaginal intercourse (79%) when compared to other participants (64%), χ^2^ = 6.21, *p* < 0.05. Because of these limited group differences, we considered all participants in all analyses and did not control for attraction.

### 3.2. Descriptive Information and Gender Differences

Table 1 presents descriptive statistics (*M*s and *SD*s) for young women and men separately, and summarizes the results of independent groups *t*-tests used to test gender differences. Young women reported a higher level of T1 sense of entitlement to sexual partner pleasure than young men. No other significant gender difference was found.

**Table 1 behavsci-06-00004-t001:** Entitlement to sexual partner pleasure, sexual self-efficacy, and sexual behavior repertoire means and SDs for young women and men, and gender comparisons (*N* = 295).

Sexual Subjectivity	Time	Total *M* (*SD*)	Young Men *M* (*SD*)	Young Women *M* (*SD*)	*t* (293)
Entitlement to partner pleasure	T1	3.88 (0.59)	3.79 (0.60)	3.94 (0.57)	−2.19 *
	T2	3.98 (0.58)	3.86 (0.58)	4.06 (0.57)	−1.93
Self-efficacy in pleasure	T1	3.71 (0.62)	3.78 (0.57)	3.66 (0.65)	1.52
	T2	3.74 (0.70)	3.81 (0.68)	3.69 (0.70)	1.08
Sexual behavior repertoire	T1	7.36 (2.31)	7.45 (2.19)	7.31 (2.38)	0.49

Notes: Young men *n* = 112; Young women *n* = 183; * *p* < 0.05, ** *p* < 0.01. *SD*s are based on the subset of 173 participants with complete data.

### 3.3. Sexual Subjectivity, Gender, Age and Sexual Behavior

Pearson correlations were computed to examine the association between T1 and T2 measures of entitlement and efficacy, sexual behavior repertoire, and age (see Table 2). At T1 and T2, the two elements of sexual subjectivity had moderate correlations with each other. Sexual behavior repertoire had positive associations with both elements of sexual subjectivity at T1 and T2 (*r*s ranged from 0.25 to 0.31, all *p* < 0.01). T1 age was associated with a greater sense of entitlement to sexual partner pleasure at T1, but age was not significantly associated with T2 entitlement or efficacy. The same pattern of correlations was found when examined separately for young men and young women.

**Table 2 behavsci-06-00004-t002:** Pearson’s correlations between the two sexual subjectivity subscales, sexual behavior repertoire and age (*N* = 295).

Variable	1	2	3	4	5
1. Entitlement to partner pleasure T1	-				
2. Self-efficacy in pleasure T1	0.43 **	-			
3. Entitlement to partner pleasure T2	0.59 **	0.26 **	-		
4. Self-efficacy in pleasure T2	0.32 **	0.62 **	0.39 **	-	
5. Sexual behavior repertoire T1	0.29 **	0.31 **	0.25 **	0.30 **	-
6. Age T1	0.14 *	0.01	0.09	0.08	0.18 *

Notes: * *p* < 0.05; ** *p* < 0.01.

### 3.4. Prospective Associations of Age, Gender, and Sexual Behavior with Entitlement and Efficacy

Two hierarchical multiple regression analyses were conducted to examine the unique associations of gender, age and sexual behavior repertoire with each element of sexual subjectivity measured at T2 (see Table 3). In each model, T2 sexual subjectivity was the dependent variable. Independent variables included gender, age, sexual behavior repertoire, and the corresponding T1 sexual subjectivity element. T1 sexual subjectivity (entitlement or efficacy), gender and T1 age were entered at Step 1. Sexual behavior repertoire was entered at Step 2 to determine whether its addition improved the prediction of sexual subjectivity over and above earlier subjectivity, age and gender alone. In Step 2 of the models, sexual behavior repertoire was positively associated with greater T2 entitlement and efficacy (relative to T1, see Table 3). Thus, participants who reported a greater variety of sexual experience at T1 reported greater increases in sense of entitlement to sexual partner pleasure and self-efficacy in achieving sexual pleasure at T2 relative to T1. Additional analyses building on these models revealed no significant gender or age interactions with sexual behavior repertoire.

**Table 3 behavsci-06-00004-t003:** Summary of results from regressing T2 sexual subjectivity on T1 sexual subjectivity, gender, age and sexual behavior repertoire (*N* = 295).

	IV	∆*R*^2^	*B* (*SE B*)	β
T2 entitlement to sexual partner pleasure				
Step 1	T1SS	0.36 **	0.57 (0.06)	0.57 **
	Gender		0.12 (0.09)	0.10
	Age		0.00 (0.02)	0.01
Step 2	T1SS	0.02 *	0.54 (0.06)	0.55 **
	Gender		0.12 (0.09)	0.10
	Age		0.00 (0.02)	−0.01
	Sexual BR		0.04 (0.02)	0.12 *
T2 sexual self-efficacy				
Step 1	T1SS	0.39 **	0.69 (0.09)	0.61 **
	Gender		−0.04 (0.09)	−0.02
	Age		0.03 (0.02)	0.08
Step 2	T1SS	0.02 *	0.65 (0.10)	0.58 **
	Gender		−0.04 (0.09)	−0.02
	Age		0.02 (0.02)	0.06
	Sexual BR		0.05 (0.02)	0.14 **

*Note*. *T1SS* = Corresponding element of Sexual subjectivity at T1; IV = independent variable; BR = behavior repertoire; * *p* < 0.05, ** *p* < 0.01; Entitlement: Step 1 *F*(3,291) = 54.6, *p* < 0.01, Final *F*(4,290) = 42.3, *p* < 0.01; Efficacy: Step 1 *F*(3,291) = 63.4, *p* < 0.01, Final *F*(4,290) = 49.8, *p* < 0.01.

### 3.5. Entitlement and Efficacy as a Function of Sexual Behavior Group and Gender

To understand the function of sexual behavior groups on sexual subjectivity, two mixed factorials ANOVAs were estimated. Sexual subjectivity (entitlement or efficacy) at T1 and T2 was entered as a within-subject factor. T1 sexual behavior group and gender were entered as between-subject factors. The ANOVA results are presented in Table 4. *M*s and *SD*s within each sexual behavior group are presented in Table 5.

**Table 4 behavsci-06-00004-t004:** Comparisons of entitlement to sexual partner pleasure and self-efficacy in achieving pleasure from T1 to T2 by sexual behavior group and gender (*N* = 295).

	Between Subject Variables, *F*			Within Subject Variables, *F*			
Dependent variable	Sex Beh (A)	Gender (B)	A × B	Time (C)	A × C	B × C	A × B × C
Entitlement to partner pleasure	4.78 **	9.60 **	1.13	12.33 **	1.56	1.83	1.03

Self-efficacy in pleasure	7.14 **	0.14	0.79	0.76	0.70	0.70	0.63


*Note*s: Sex Beh = sexual behavior group; ** *p* < 0.01.

**Table 5 behavsci-06-00004-t005:** Mean and SD of entitlement to sexual partner pleasure and self-efficacy in achieving pleasure by sexual behavior group (*N* = 295).

		Sexual Experience Group, *M* (*SD*)				
Dependent variable	Time	I (*n* = 27)	E (*n* = 47)	C > 16 (*n* =108)	C16 (*n* = 53)	C < 16 (*n* = 60)
Entitlement to partner pleasure	T1	3.51 (0.81)	3.78 (0.54)	3.70 (0.54)	4.03 (0.52)	3.97 (0.59)
	T2	3.68 (0.69)	3.65 (0.57)	3.98 (0.52)	3.90 (0.62)	3.79 (0.56)
Self-efficacy in pleasure	T1	3.24 (0.77)	3.65 (0.64)	3.70 (0.59)	3.90 (0.40)	3.79 (0.64)
	T2	3.26 (0.70)	3.61 (0.70)	3.79 (0.64)	3.90 (0.62)	3.82 (0.76)

*Note*: I = inexperienced; E = experienced no coitus; C > 16 = first sexual intercourse after age 16; C16 = first sexual intercourse at age 16; C < 16 = first sexual intercourse before age 16.

For sense of entitlement to sexual partner pleasure, significant main effects were found for sexual behavior, gender, and time (see Table 4). Pairwise comparisons revealed that inexperienced participants reported less entitlement when compared to the three groups that reported a history of coitus (see Table 5 for means and *SD*s within sexual behavior groups). Young women reported more sense of entitlement to sexual partner pleasure than young men, and the average level of entitlement increased from T1 to T2 (see Table 1).

For sexual self-efficacy, a significant main effect was found for sexual behavior group (see Table 4 and Table 5). Pairwise comparisons revealed that inexperienced participants reported less self-efficacy in achieving sexual pleasure than the three groups that reported a history of coitus.

## 4. Discussion

The current study was the first of its kind to illustrate how aspects of sexual subjectivity, assessed as a sense of entitlement to sexual partner pleasure and self-efficacy to achieve sexual pleasure, progressed over one year in *both* young women and men (aged 17–25 years). We also investigated whether entitlement and efficacy differed between young men and young women, and if they were associated with age and past sexual experience. In general, there were stronger associations of sexual experiences with sexual subjectivity, when compared to the associations of gender and age with sexual subjectivity. Those participants with relatively more diverse sexual behavior histories reported greater increases in both their entitlement and efficacy from T1 to T2, and inexperience in sexual behavior clearly differentiated a group with lower entitlement and efficacy when compared to those with a history of coital experience.

### 4.1. Sexual Subjectivity as a Correlate of Gender, Age, and Sexual Behavior

Only recently was a measure developed to assess sexual subjectivity in both young men and women made available [53]. Therefore, this was the first study to explore gender differences in levels of sexual subjectivity, and examine change in sexual subjectivity over time in both young men and women. When a sense of entitlement to sexual partner pleasure and sexual self-efficacy were compared, one gender difference was found. Young women felt more entitled to sexual partner pleasure at the start of the study (Time 1) when compared to young men. This finding compliments the broader research on gender difference in sexual development and behavior [38], which indicates young women place more emphasis than young men on the role of a sexual partner to meet their sexual desires and pleasure.

It was unexpected that no significant gender difference was found when comparing self-efficacy in achieving pleasure. In a previous cross-sectional study, young women had significantly lower levels of self-efficacy compared to men [53]. However, this finding is consistent with the other research that identifies the preponderance of gender similarity over difference in general sexual perceptions and attitudes [43].

Other aims of the current study were to examine associations of sexual subjectivity with age and sexual behavior. In relation to age, it was associated with a greater sense of entitlement to sexual partner pleasure at T1. In other research on young women's sexual subjectivity, the findings have been mixed regarding the association between age and sexual subjectivity. Some studies have found no association between age and sexual subjectivity (e.g., [34]). However, other research has found differences in sexual subjectivity when age groups are compared (e.g., age 20 and older to under 20) [19]. Taken together, these results suggest that age may play only a small role in the progression of sexual subjectivity, and it may not be incremental age that is most revealing of associations with sexual subjectivity. The disparity in past and present research highlights the need for future research to remain focused on understanding the typical development of sexual subjectivity, before shifting its focus on to atypical development [27]. It may be that a momentous relationship change (e.g., a first commitment to what is expected to be a lifelong partner), which is more likely during the early adult years, may be the fuel for a leap in sexual subjectivity and comparing age groups before or after such a change might reveal the greatest age difference. However, of course, this still suggests that it is not age per se that is associated with sexual subjectivity, rather it is experiences that become more common with increasing age that are most relevant.

The above stresses our view about the importance of experience for sexual development. In the present study, experience was examined as sexual behavior history and this was investigated through two different avenues. First, the range of experience with different sexual behaviors was measured as the repertoire of sexual experience and examined as a correlate of entitlement and efficacy. Second, five groups were formed based on sexual behavior history. These groups ranged from participants who were quite inexperienced with sexual behavior to participants who reported first coitus the earliest (before the age of 16). Overall, as expected, a sense of entitlement to sexual partner pleasure and sexual self-efficacy were heightened among those who reported a more varied history or earlier onset of sexual behavior. In particular, it was especially the participants with little sexual experience who were significantly lower in a sense of entitlement to sexual partner pleasure and efficacy to achieve pleasure.

### 4.2. Sexual Subjectivity Patterns Over Time

Our findings show that entitlement to sexual partner pleasure among those aged 17 to 25 years does increase even across only a 1-year period. Also, at the beginning of the study, young women reported more sense of entitlement than young men, and older participants reported more entitlement relative to younger participants. However, we found no evidence that the average pattern of increase in entitlement differed between young men and women or differed between groups differentiated by their sexual behavior histories. When all of these findings are considered, they suggest that young women may be slightly more advanced than young men (and those in their 20 s may be more advanced than those in their teens) in their sense of entitlement, but that, on average, young people in our sample tended to increase during the late teens and early 20 s. This also suggests that young people may start to converge on a similar level of entitlement resulting in less difference by gender and age over time.

Surprisingly, the same pattern was not found for participants' reports of their efficacy to achieve sexual pleasure. Thus, although entitlement and efficacy are moderately correlated with each other, it seems to be entitlement that is progressing in the late teens and early 20 s and not efficacy. Efficacy also tended to be lower, on average, than entitlement. Moreover, efficacy was higher with more sexual behavior experience. We hypothesize that efficacy develops later after sexual relationships become more stabilized and committed. Future research seems needed that can follow participants over a longer period of time to examine this possibility.

### 4.3. Study Limitations and Future Directions

Although this study provided insight into how aspects of sexual subjectivity differed over one year in young men and women and uncovered associations of age and sexual behavior with sexual subjectivity, there were two limitations worthy of note. First, the participants in the study were predominantly middle class, Caucasian students at one university, who were residing in one region of Australia. There is also the possibility that individuals more interested in sex and relationships chose to participate. Therefore, the results of the current study may have some limited generalizability. Second, probably because the majority of the participants had already experienced coitus prior to the first data collection and because we measured types of sexual behavior and not frequency or some other aspect of behavior, there was high stability in sexual behavior over the one year of this study. Statistics show that the majority of Australian year 10–12 students (approximately 15–17 years) have engaged in some form of sexual behavior [12]. Therefore, in order to capture sexual subjectivity as it emerges and develops most rapidly along with sexual behavior, future studies may need to begin with a younger population. 

Recently, Hensel *et al.* [31] found that, in their adolescent participants, sexual self-concept had a bidirectional relationship with sexual experience, and the same may occur for sexual subjectivity. Future research could examine the possibility of bidirectional relationships between sexual subjectivity and sexual behavior. In addition, no previous study has examined the timing of pubertal development and sexual subjectivity. Evidence suggests that adolescents that mature earlier, compared to their peers, form romantic relationships earlier and experience sexual behaviors earlier [11,55]. In the current and past research, the link between age and sexual subjectivity has been relatively weak [34]. Substituting timing of pubertal status for age may provide more evidence regarding individual characteristics that influence differences in sexual subjectivity and its development over time.

## 5. Conclusions

Sexual subjectivity has been considered an aspect of sexual health partly because it has been associated with greater positive well-being in empirical research [19,34]. For example, elevated sexual subjectivity has been associated with greater overall positive psychosocial functioning in young women [19,24,34] and young men [53]. More specifically, it has been demonstrated that sexual subjectivity was related to adolescents' greater endorsement of their condom use self-efficacy, sexual esteem, global self-esteem, and global well-being [19,53]. In addition, elevated sexual subjectivity was related to lower levels of sexual depression and sexual anxiety [19]. Finally, sexual self-concept and well-being have been examined over time. In one study, Cheng, Hamilton, and Massari [56] explored the long term consequences of two components of sexual subjectivity in adolescent girls. Using data collected through the USA National Longitudinal study of Adolescent Health, they examined teenage girls’ expectations of pleasure during intercourse and sexual self-efficacy. They found that these two components predicted young adult well-being in sexual, mental and physical health. Whereby, girls with higher expectations for pleasure during intercourse and higher levels of sexual self-efficacy were more likely to report more consistent condom use and to avoid teenage pregnancy. In addition, girls with higher sexual self-efficacy reported significantly better mental and physical health in adulthood. Overall, these findings provide support for Tolman’s [29] theory that sexual subjectivity assists young women to be self-motivated actors and to make responsible choices about their sexual behavior. Given the little difference found between young women and men in the current study, it now seems relevant to investigate whether sexual subjectivity in young men plays some role in the development of their well-being, relationship formation and maintenance, sexual satisfaction, and sexually protective behaviors. It also is equally important to consider the social environment within which young women and men engage in sexual behavior. These environments continue to be marked by stereotypes, expectations, and social power differentials between men and women that may render greater gender differences in sexual subjectivity and other aspects of the sexual self-concept than we have found in the present study [22,57,58,59].

There are multiple dimensions of behaviors, cognitions, attitudes and emotions that contribute to the development of sexuality [26,59]. The study of sexual subjectivity provides insight into the sexual self-perceptions of entitlement to sexual partner pleasure and efficacy in achieving it; both are important aspects of understanding oneself as a sexual person [25,30]. Both young men and women seem to increase in sexual subjectivity as they explore and gather more experience with a range of sexual behaviors. Moreover, although young men and women do sometimes differ in their levels of sexual subjectivity, these differences are relatively small and not consistent across studies. This accumulating information about sexual subjectivity, especially entitlement to sexual partner pleasure and sexual self-efficacy, in young men and women, as well as the growing body of theory and research on sexual health beyond its lack of sexual problems [3,15,17,18,19,20,21,22,23,24,25,26,34,60], provides researchers, clinicians and other practitioners and sexual health experts alike, the opportunity to draw from this research to more clearly articulate and promote optimal sexual health.

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
