# Peer review of "A Longitudinal Study of Sexual Entitlement and Self-Efficacy among Young Women and Men: Gender Differences and Associations with Age and Sexual Experience"

_behavsci, 2016, doi:10.3390/bs6010004_

Round 1

Reviewer 1 Report

There might be some interesting data collected in the present study.  However, my main concern with the study is that there were too many analyses presented to interpret in a meaningful way.  The authors should revise the paper providing more theoretical background and accompanying analyses.   With better theoretical framing and a more limited set of analyses, the value of the study results should be more clear.  Some more specific suggestions are provided below.

I think it would be preferable to describe the measurement of sexual subjectivity shortly after it is defined on p. 3.  Also, a more thorough description of the measurement of the construct and the validity of the measurement could be described in that section.

On p. 4 the first sentence of second full paragraph, “Regarding sexual subjectivity and its link to greater sexual exploration and experience, one cross-sectional with a sample of females” is missing the word “study.”

The authors could provide more theoretical explanation of why there might be gender differences in sexual subjectivity. 

The authors rely too much on headings to transition between ideas.  The paper would flow more smoothly if some transition sentences were included.

More specific hypotheses could be proposed.  The section where research questions are described is worded in an awkward way.

The reporting of group differences in the participants section might be better placed in the results section after the measures are described.  

The combining of the male and female versions of the sexual subjectivity inventories should be described in more detail, and the validity of creating the subscales should be explained.  Perhaps factor analysis could be used to determine appropriate subscales?

What were participants told about the purpose of the study?

There is an increase in Type I error with the multiple t-tests conducted thatshould be addressed.  Similarly, there are many correlation tests performed which increases Type I error that needs to be addressed.

Predictions about how the specific subscales of the inventory might relate to age and experience might be provided to make the results more meaningful (see the comment about theory relating to analyses discussed above).

Author Response

There might be some interesting data collected in the present study.  However, my main concern with the study is that there were too many analyses presented to interpret in a meaningful way.  The authors should revise the paper providing more theoretical background and accompanying analyses.   With better theoretical framing and a more limited set of analyses, the value of the study results should be more clear.  Some more specific suggestions are provided below.

***Thank you for this advice. We have trimmed the paper to focus on two of the five elements of sexual subjectivity - only those that were expected to be most relevant to partnered sexual partner: entitlement to sexual partner pleasure and efficacy in achieving pleasure. This has substantially limited the analyses conducted. We provide a rationale for this (see the first paragraph on p.4) and have thoroughly revised the entire paper.

I think it would be preferable to describe the measurement of sexual subjectivity shortly after it is defined on p. 3.  Also, a more thorough description of the measurement of the construct and the validity of the measurement could be described in that section.

***We have done this and have also substantially revised the entire Introduction (see new section starting on p. 3 titled Definition and measurement).

On p. 4 the first sentence of second full paragraph, “Regarding sexual subjectivity and its link to greater sexual exploration and experience, one cross-sectional with a sample of females” is missing the word “study.”

**We have corrected this sentence.

The authors could provide more theoretical explanation of why there might be gender differences in sexual subjectivity. 

***We have added to our theoretical explanation for why there would be gender differences (see the section titled Gender Differences in Sexual Behavior, Attitudes, and Sexual Subjectivity, p. 4).

The authors rely too much on headings to transition between ideas.  The paper would flow more smoothly if some transition sentences were included.

***We have substantially revised all sections of this paper to improve the flow and removed some headings.

More specific hypotheses could be proposed.  The section where research questions are described is worded in an awkward way.

***We have revised this section (The Current Study, p.5). We have also added hypotheses (see p. 5-6)

The reporting of group differences in the participants section might be better placed in the results section after the measures are described.  

***We have moved this content on group difference to the first section of the Results section.

The combining of the male and female versions of the sexual subjectivity inventories should be described in more detail, and the validity of creating the subscales should be explained.  Perhaps factor analysis could be used to determine appropriate subscales?

***Given our focus on only two element of sexual subjectivity, this was no longer an issue because items were the same on measures for males and females.

What were participants told about the purpose of the study?

***We have added this information on p. 7 in the Procedure section. It reads:

Approval from the university Human Research Ethics Committee was obtained prior to data collection. At Time 1 (T1), participants were approached at a university campus in Australia in the week before classes commenced (i.e., during orientation week) and asked to participate in a study "About You and Your Relationships." The front cover of the survey described the questions as focused on personal sexual and romantic experiences, and stressed the confidential nature of the survey.

There is an increase in Type I error with the multiple t-tests conducted thatshould be addressed.  Similarly, there are many correlation tests performed which increases Type I error that needs to be addressed.

 ***Reducing the number of elements of sexual subjectivity has reduced the number of tests performed.

Predictions about how the specific subscales of the inventory might relate to age and experience might be provided to make the results more meaningful (see the comment about theory relating to analyses discussed above).

***We have added hypotheses to the section titled The Current Study on p. 5-6.

Reviewer 2 Report

This paper presents results from a longitudinal study of sexual subjectivity in university-age students who were followed over the course of 1 year. The normative development of sexual health is an important and understudied topic, and the Introduction to this paper provides a good overview of the theoretical work in this area. I have four recommendations for improving the paper.

First, the attrition between Time 1 and Time 2 was substantial (40% of participants), and this is handled using list-wise deletion. That is, any one who did not complete both assessments was not included in the analyses. List-wise deletion can lead to serious bias in results. Analyses should be re-done using a more sophisticated method for handling missing data, such as multiple imputation or full-information maximum likelihood (Shafer & Graham, 2002, Psychological Methods). 

Second, potential participants were "approached at a university campus" during orientation week. What percentage of people who were approached agreed to participate? How broadly representative of the university population (or the university-age population) is the sample? From the relatively high percentage of non-heterosexual participants, there seems to be some selection bias. At the very least, the Discussion should consider how generalizable these results are.

Third, although longitudinal data can be a quite powerful tool for understanding change and development, the current analyses squander some of that power. The regression results presented in Table 2 test whether sexual behaviors that participants already experienced by Time 1 predicted facets of sexual subjectivity at Time 2, controlling for sexual subjectivity at Time 1. Why would behavior that has already happened contribute to a re-ordering of individuals over the course of the next year? It seems that a more interesting and direct test of the longitudinal effects of sexual behavior on sexual subjectivity would test whether new sexual behaviors (that is, sexual behaviors experienced between Time 1 and Time 2) predict change in sexual subjectivity from Time 1 to Time 2. Moreover, the reverse paths -- from sexual subjectivity to future behavior -- are also not tested in this paper. More generally, I think the putative timescale of effects of behavior on sexual subjectivity needs more theoretical elaboration. Are behaviors in early adolescence (before age 16), for example, expected to continue to influence the trajectory of sexual subjectivity regardless of later sexual behavior? That seems to be the model implied by the analyses, but it is never explicitly specified.

Fourth, each analysis is conducted for each subscale separately, but (with the exception of sexual body-esteem) the scales are consistently (if moderately) intercorrelated. I think it would be informative to test whether associations with age or sexual experience group are operating through a general underlying factor of sexual subjectivity versus are unique to specific facets. The shift to a latent factor approach would have the added benefit of allowing the authors to use FIML to account for missing data at T2, and they could thus use their entire sample. 

Fifth, one set of results seems quite contradictory: sense of entitlement to sexual self-pleasure decreased, on average, from Time 1 to Time 2, but sense of entitlement to sexual self-pleasure was positively correlated with age. How do the authors make sense of these result, as participants are one year older at Time 2? This contradiction underscores my concern about attrition (point #1); I wonder if the seeming decrease from Time 1 to Time 2 is an artifact of attrition.

Finally, on a selfish note, I wrote an extensive review piece on this topic (Harden, 2014, "A Sex-Positive Framework for Reseach on Adolescent Sexuality" in Perspectives on Psychological Science), and I would, of course, be happy if the authors included a reference to my paper in their Introduction or Discussion.

Author Response

This paper presents results from a longitudinal study of sexual subjectivity in university-age students who were followed over the course of 1 year. The normative development of sexual health is an important and understudied topic, and the Introduction to this paper provides a good overview of the theoretical work in this area. I have four recommendations for improving the paper.

***Thank you for the compliments and the suggestions.

First, the attrition between Time 1 and Time 2 was substantial (40% of participants), and this is handled using list-wise deletion. That is, any one who did not complete both assessments was not included in the analyses. List-wise deletion can lead to serious bias in results. Analyses should be re-done using a more sophisticated method for handling missing data, such as multiple imputation or full-information maximum likelihood (Shafer & Graham, 2002, Psychological Methods). 

***We have now used multiple imputation to maintain all 295 participants in all analyses.

Second, potential participants were "approached at a university campus" during orientation week. What percentage of people who were approached agreed to participate? How broadly representative of the university population (or the university-age population) is the sample? From the relatively high percentage of non-heterosexual participants, there seems to be some selection bias. At the very least, the Discussion should consider how generalizable these results are.

***We are sorry that we did not keep a tally of the number of students approached.  We have added to the Limitations section of the Discussion about this (see p. 12). It reads,

First, the participants in the study were predominantly middle class, Caucasian university students, residing in one region of Australia. There is also the possibility that individuals more interested in sex and relationships chose to participate. Therefore, the results of the current study may have some limited generalizability.

Third, although longitudinal data can be a quite powerful tool for understanding change and development, the current analyses squander some of that power. The regression results presented in Table 2 test whether sexual behaviors that participants already experienced by Time 1 predicted facets of sexual subjectivity at Time 2, controlling for sexual subjectivity at Time 1. Why would behavior that has already happened contribute to a re-ordering of individuals over the course of the next year? It seems that a more interesting and direct test of the longitudinal effects of sexual behavior on sexual subjectivity would test whether new sexual behaviors (that is, sexual behaviors experienced between Time 1 and Time 2) predict change in sexual subjectivity from Time 1 to Time 2. Moreover, the reverse paths -- from sexual subjectivity to future behavior -- are also not tested in this paper. More generally, I think the putative timescale of effects of behavior on sexual subjectivity needs more theoretical elaboration. Are behaviors in early adolescence (before age 16), for example, expected to continue to influence the trajectory of sexual subjectivity regardless of later sexual behavior? That seems to be the model implied by the analyses, but it is never explicitly specified.

*** We agree that this is very important. We have added the Introduction to expand on these points. Unfortunately we could not conduct these suggested analyses. We had very little change in sexual behavior given the way we measured it. Also, we did not have confidence in estimation of sexual behavior for those who did not participate at Wave 2, which could have increased the change.  Thus, we have maintained our original analyses, but make this comment in the Discussion (see p. 12-13):

Second, probably because the majority of the participants had already experienced coitus prior to the first data collection and because we measured types of sexual behavior and not frequency or some other aspect of behavior, there was high stability in sexual behavior over the one year of this study. Recent statistics show that the majority of Australian year 10-12 students (approximately 15-17 years) have engaged in some form of sexual behavior [12]. Therefore, in order to capture sexual subjectivity as it emerges and develops most rapidly along with change in sexual behavior, future studies may need to begin with a younger population.

Recently, Hensel et al. [31] found that, in their adolescent participants, sexual self-concept had a bidirectional relationship with sexual experience, and the same may occur for sexual subjectivity. Future research could examine the possibility of bidirectional relationships between sexual subjectivity and sexual behavior. In addition, no previous study has examined the timing of pubertal development and sexual subjectivity. Evidence suggests that adolescents that mature earlier, compared to their peers, form romantic relationships earlier and experience sexual behaviors earlier [11,55]. In the current and past research, the link between age and sexual subjectivity has been relatively weak [34]. Substituting timing of pubertal status for age may provide more evidence regarding individual characteristics that influence differences in sexual subjectivity and its development over time.

Fourth, each analysis is conducted for each subscale separately, but (with the exception of sexual body-esteem) the scales are consistently (if moderately) intercorrelated. I think it would be informative to test whether associations with age or sexual experience group are operating through a general underlying factor of sexual subjectivity versus are unique to specific facets. The shift to a latent factor approach would have the added benefit of allowing the authors to use FIML to account for missing data at T2, and they could thus use their entire sample. 

***We have trimmed the paper to focus on two of the five elements of sexual subjectivity and anticipated that these two elements would show different results. Thus, we have maintained the separate analyses of these two elements. Please see the added Hypotheses on p. 5-6.

Fifth, one set of results seems quite contradictory: sense of entitlement to sexual self-pleasure decreased, on average, from Time 1 to Time 2, but sense of entitlement to sexual self-pleasure was positively correlated with age. How do the authors make sense of these result, as participants are one year older at Time 2? This contradiction underscores my concern about attrition (point #1); I wonder if the seeming decrease from Time 1 to Time 2 is an artifact of attrition.

***We have removed these analyses from the paper in an attempt to reduce the number of analyses and streamline the entire paper.

Finally, on a selfish note, I wrote an extensive review piece on this topic (Harden, 2014, "A Sex-Positive Framework for Reseach on Adolescent Sexuality" in Perspectives on Psychological Science), and I would, of course, be happy if the authors included a reference to my paper in their Introduction or Discussion.

***Thank you for this suggestion. The reference has been added to the paper.

Round 2

Reviewer 1 Report

The authors did a nice job of addressing the concerns I pointed out in my previous review.   The paper is more streamlined and focused, and the research findings presented are novel.

Author Response

Please specify how many students were approached to participate in the study and therefore specify the response rate. A low response rate will seriously limit the validity of the findings and this should be discussed as an important limitation also.

**We did not keep an exact count of the number of students approached about the study, but we estimate that we approached about 375 students. We have added this information to the paper.

Important limitations of the study including that this was a study of students at one university only should be included in the Abstract.

** We added a comment that data were collected at one Australian university to the abstract, and added a summary statement and note about limitations related to generalizability.

The abstract now reads:

Abstract: Many scholars have called for an increased focus on positive aspects of sexual health and sexuality. Using a longitudinal design with two assessments, we investigated patterns of entitlement to sexual partner pleasure and self-efficacy to achieve sexual pleasure among 295 young men and women aged 17-25 years attending one Australian university. We also tested whether entitlement and efficacy differed by gender, and hypothesized that entitlement and efficacy would be higher in older participants and those with more sexual experience. A sense of entitlement to sexual partner pleasure increased significantly over the year of the study, whereas, on average, there was no change in self-efficacy over time. At Time 1 (T1), young women reported more entitlement than young men. Age was positively associated with T1 entitlement, and experience with a wider range of partnered sexual behaviors was concurrently associated with more entitlement and efficacy and was also associated with increased entitlement to partner pleasure and increased self-efficacy in achieving sexual pleasure at T2 relative to T1. A group with the least amount of sexual experience was particularly low in entitlement and efficacy when compared to groups with a history of coital experience. There was no evidence that any association differed between young men and young women. Limitations of the study include a sample of predominantly middle class, Caucasian students at one university and the possibility that students more interested in sex and relationships, and with more sexual experience, chose to participate.

**We also expanded the first paragraph of the Limitations to say:

            Although this study provided insight into how aspects of sexual subjectivity differed over one year in young men and women and uncovered associations of age and sexual behavior with sexual subjectivity, there were two limitations worthy of note. First, the participants in the study were predominantly middle class, Caucasian students at one university, who were residing in one region of Australia. There is also the possibility that individuals more interested in sex and relationships chose to participate. Therefore, the results of the current study may have some limited generalizability. Second, probably because the majority of the participants had already experienced coitus prior to the first data collection and because we measured types of sexual behavior and not frequency or some other aspect of behavior, there was high stability in sexual behavior over the one year of this study. Statistics show that the majority of Australian year 10-12 students (approximately 15-17 years) have engaged in some form of sexual behavior [12]. Therefore, in order to capture sexual subjectivity as it emerges and develops most rapidly along with sexual behavior, future studies may need to begin with a younger population.

Moreover there is a comment in the Methods that 80% of the subjects reported being only attracted to members of the same sex. This seems to contrast with later statements and would also seem to be an unlikely finding. Please address this matter.

**Thank you catching the typographical error (80% attracted to the same sex) - this should have read "other sex".

The first paragraph of the Participants section now reads:

The participants were 295 adolescent and young adult men (n = 112) and women (n = 183) aged between 17 to 25 years (M = 19.5 years, SD = 1.9). Although an exact count was not available, we estimate that 375 students were approached to participate, resulting in a participation rate of approximately 79%. Participants completed two waves of data collection approximately 1-year apart. Overall, 91% were white/Caucasian, 3% were Asian, 1 was Aboriginal/Pacific Islander, and the remaining participants indicated an "Other" sociocultural background. Most lived with their parents (61%) and reported being only attracted to the other sex (80%). At T1, 225 of the participants (76%) reported a history of sexual intercourse and 9% reported having had no sexual experience apart from kissing.

Please note - we have also shortened the title, found some other typographical errors to fix, and fixed some references.